# Acupuncture Decreases Risk of Hypertension in Patients with Chronic Spontaneous Urticaria in Taiwan: A Nationwide Study

**DOI:** 10.3390/healthcare11101510

**Published:** 2023-05-22

**Authors:** Heng-Wei Chang, Wei-De Lin, Pai-Jun Shih, Shin-Lei Peng, Chung-Y. Hsu, Cheng-Li Lin, Wen-Ling Liao, Mao-Feng Sun

**Affiliations:** 1Graduate Institute of Chinese Medicine, China Medical University, Taichung 404333, Taiwan; handway@gmail.com; 2Department of Medical Research, China Medical University Hospital, Taichung 404327, Taiwan; weide@mail.cmuh.org.tw; 3Erlin Four Seasons Dermatology Clinic, Erlin Township, Changhua County 526022, Taiwan; byron.shih@gmail.com; 4Department of Biomedical Imaging and Radiological Science, China Medical University, Taichung 404333, Taiwan; 5Graduate Institute of Biomedical Sciences, China Medical University, Taichung 404333, Taiwan; 6Management Office for Health Data, China Medical University Hospital, Taichung 404327, Taiwan; 7Graduate Institute of Integrated Medicine, College of Chinese Medicine, China Medical University, Taichung 404333, Taiwan; 8Center for Personalized Medicine, China Medical University Hospital, Taichung 404327, Taiwan; 9Graduate Institute of Acupuncture Science, College of Chinese Medicine, China Medical University, Taichung 404333, Taiwan

**Keywords:** hypertension, acupuncture, traditional Chinese medicine, chronic spontaneous urticaria

## Abstract

Patients with chronic spontaneous urticaria (CSU) have a higher risk of developing hypertension. This study aimed to determine whether acupuncture could decrease the risk of hypertension in patients with CSU. We enrolled patients newly diagnosed with CSU between 1 January 2008, and 31 December 2018, from the Taiwanese National Health Insurance Research Database. The claims data were assessed from the index date to 31 December 2019. A Cox regression model was used to compare the hazard ratios (HRs) of the two cohorts. The cumulative incidence of hypertension was estimated using the Kaplan–Meier method. After propensity score matching with a 1:1 ratio, 43,547 patients with CSU who received acupuncture were matched with 43,547 patients with CSU who did not receive acupuncture in this study. After considering potential confounding factors, patients who received acupuncture had a significantly lower risk of hypertension than those in the control group (adjusted hazard ratio = 0.56, 95% confidence interval = 0.54–0.58). Patients who received medications combined with acupuncture tended to have the lowest risk of hypertension. This study revealed that acupuncture decreases the risk of hypertension in patients with CSU in Taiwan. The detailed mechanisms can be further clarified through prospective studies.

## 1. Introduction

Hypertension is a disease that is highly prevalent worldwide and increases the risk of several fatal conditions, such as ischemic stroke, intracerebral hemorrhage, acute myocardial infarction, heart failure, and aortic dissection [1,2,3]. Hypertension is the most prevalent modifiable risk factor for premature cardiovascular disease, even when compared to cigarette smoking, dyslipidemia, and diabetes [4]. The exact etiology and mechanism of primary hypertension remain unclear, but the risk factors, such as age, obesity, family history, race, and a high-sodium diet, are well-known [5]. The more risk factors we become aware of, the better we can prevent or control hypertension.

A previous study found that chronic idiopathic urticaria (CIU) was associated with a higher risk of hypertension [6]. This study reminded us that patients with CIU may be considered for the evaluation of masked hypertension. CIU is an intractable skin disease characterized by persistent intense itching, wheals, and/or angioedema lasting for six weeks or longer, as opposed to acute urticaria, which does not have any specific identifiable triggers. The terms “chronic urticaria,” “chronic idiopathic urticaria,” and “chronic spontaneous urticaria” have been used interchangeably, but “chronic spontaneous urticaria,” or CSU, has been most widely used by specialists [7], as it emphasizes an endogenous cause and non-inducible nature. Another study reported that hypertension is associated with an extended duration of CSU [8]. However, CSU and hypertension, which belong to different medical specialties, have rarely been discussed together in modern medicine.

Acupuncture has been widely used in Asian [9,10,11] and Western countries [12,13,14,15]. In Taiwan, the National Health Insurance (NHI) program provides coverage for more than 23 million residents. Traditional Chinese medicine (TCM) services, including acupuncture, have been reimbursed since 1996 [16]. The Taiwanese National Health Insurance Research Database (NHIRD) includes demographic, diagnostic, interventional, and long-term follow-up data on more than 99% of the population in Taiwan [17]. A study found that 23% of people in Taiwan used acupuncture from 1996 to 2002 [9]. Another study revealed that acupuncture was increasingly popular in Taiwan from 2002 to 2011 [18]. Acupuncture is often used in combination with Western medications. To determine whether acupuncture could decrease the risk of hypertension in patients with CSU, we used the NHIRD to conduct a population-based cohort study.

The study was conducted between 2008 and 2019 and included patients with CSU who received acupuncture or not. The primary endpoint was the occurrence of hypertension. The two cohorts were matched through 1:1 propensity score matching by age, sex, index year, comorbidities, medications, monthly income, urbanization level, and occupation. The study included 43,547 patients with CSU receiving acupuncture matched with 43,547 patients with CSU who did not receive acupuncture. The results showed that the acupuncture group had a significantly lower risk of hypertension than those in the control group (adjusted HR = 0.58, 95% confidence interval (CI) = 0.55–0.60). In the interaction analysis, patients receiving medications combined with acupuncture tended to have the lowest risk of hypertension.

## 2. Materials and Methods

### 2.1. Data Source

We used data from the NHIRD from January 1, 2008, to December 31, 2019, as our data source, provided by the Health and Welfare Data Science Center (HWDC) of the Ministry of Health and Welfare of Taiwan. Healthcare information, including outpatient visits, hospital admissions, and prescription medications, was used in this study, and the International Classification of Disease, Ninth & Tenth Revision, Clinical Modification (ICD-9-CM & ICD-10-CM) recorded disease diagnoses. NHIRD provides a large sample size and reduces selection bias while including long-term follow-up data. HWDC policies ensure the safety and privacy of this data source. The institutional review board of China Medical University (CMUH110- REC1-038[CR-2]) has approved this study.

### 2.2. Study Population

The study population comprised patients with CSU (ICD 9 708.1, ICD 10 L50.1). Patients who received acupuncture between 2008 and 2018 were defined as the acupuncture cohort and those who did not receive acupuncture were defined as the non-acupuncture cohort. The index date was the first date of acupuncture in the acupuncture cohort, whereas a random date between 2008 and 2018 was the index date in the non-acupuncture cohort. We excluded patients who had missing sex or age data (1.3%) and those who had any hypertension diagnosis before the index date. In addition, the two cohorts were matched through 1:1 propensity score matching by age, sex, index year, comorbidities, medications, monthly income, urbanization level, and occupation. A flow chart for study design is shown in the Appendix A (Appendix A).

### 2.3. Primary Endpoint and Confounding Variables

The occurrence of hypertension (ICD 9 401-405, ICD 10 I10-I15) between 2008 and 2019 was considered the primary endpoint. Hypertension patients were identified at least twice from principal/secondary diagnoses in outpatient visits or one visit with hospitalizations from 2008 to 2019 during the follow-up period. All patients were followed from the index date to the occurrence of hypertension, death, withdrawal from the NHI program, or 31 December 2019. Patients with diabetes mellitus or hyperlipidemia comorbidities or those receiving cetirizine, desloratadine, fexofenadine, levocetirizine, or loratadine medications before the endpoint were included as confounding variables. In addition, patient characteristics including monthly income, urbanization level, and occupation were considered.

### 2.4. Statistical Analysis

Continuous variables of patient characteristics were presented as means and standard errors, and categorical variables were presented as numbers and percentages. The differences between the two cohorts were evaluated using Student’s *t*-test and χ^2^ test. Univariate and multivariate Cox regression models were used to estimate the crude hazard ratio (HR) and adjusted HR of hypertension associated with risk factors. The multivariate analysis included age, sex, comorbidities, medications, monthly income, urbanization level, and occupation. The competing-risk regression models accounted for the competing risk of death and computed the subhazard ratio (SHR) of hypertension. Kaplan–Meier survival curves were plotted to compare the cumulative incidence rate of hypertension, and the log-rank test was used to examine the differences between the acupuncture and non-acupuncture cohorts. All statistical analyses were performed using SAS version 9.4 (SAS Institute Inc., Cary, NC, USA) or R Studio (version 3.5.2).

## 3. Results

### 3.1. Demographic Characteristics of Study Population

Table 1 shows the 43,547 patients with CSU receiving acupuncture matched with 43,547 patients with CSU who did not receive acupuncture in this study. The average ages of the acupuncture and non-acupuncture cohorts were 40.1 ± 13.0 years and 40.2 ± 13.2 years, respectively. Approximately 68% of patients with CSU were female. The majority of patients had white-collar occupations and lived in areas of urbanization level 1. The number of patients with diabetes mellitus or hyperlipidemia did not differ between the two cohorts (*p* > 0.05). The proportions of patients with CSU, with and without acupuncture, who were on medication were as follows: cetirizine (86.1% vs. 79.2%), desloratadine (42.2% vs. 36.0%), fexofenadine (93.7% vs. 89.1%), levocetirizine (71.5% vs. 63.0%), and loratadine (72.0% vs. 65.3%).

### 3.2. Risk of Hypertension According to Whether the CSU Patients Underwent Acupuncture and Other Covariates

The risk of hypertension according to the presence of covariates is presented in Table 2. The results showed that patients treated with acupuncture had a significantly lower risk of hypertension than those in the control group (adjusted HR = 0.58, 95% confidence interval (CI) = 0.55–0.60). The Kaplan–Meier curves in Figure 1 shows a visual picture of the cumulative incidence of hypertension. Patients with comorbidities had a significantly higher risk of hypertension (adjusted HR = 1.15 and 1.51 for type 2 diabetes and hyperlipidemia, respectively, both *p* < 0.001). In contrast, patients receiving medication had a significantly lower risk of hypertension (adjusted HRs < 1, *p* < 0.001). Compared to white-collar occupation patients, blue-collar occupation patients had a significantly higher risk of hypertension (adjusted HR = 1.22, 95% CI = 1.17–1.28, *p* < 0.001), while other patients had a significantly lower risk of hypertension (adjusted HR = 0.79, 95% CI = 0.62–0.99). Patients who lived in urbanization level 2 (adjusted HR = 1.23, 95% CI = 1.03–1.46, *p* = 0.02) and level 3 (adjusted HR = 1.31, 95% CI = 1.09–1.58, *p* = 0.004) areas had a greater risk of hypertension than those who lived in urbanization level 4 areas. Moreover, patients with a monthly income lower than NTD 20,000 had a higher risk than those with a monthly income greater than NTD 40,000 (adjusted HR = 1.42, 95% CI = 1.13–1.79, *p* = 0.001).

### 3.3. Risk of Hypertension among Patients with CSU Who Received and Did Not Receive Acupuncture

Table 3 shows that regardless of sex, age, comorbidities, and medications, acupuncture users tended to have a lower risk of hypertension than non-acupuncture users (all adjusted HRs < 1, *p* <0.001). The results showed that patients treated with acupuncture had a significantly lower risk of hypertension than those without acupuncture treatment (adjusted HR = 0.52, 95% confidence interval (CI) = 0.50–0.55 for female and adjusted HR = 0.56, 95% confidence interval (CI) = 0.53–0.60 for male). Table 4 shows the SHRs of hypertension in the acupuncture and non-acupuncture cohorts. Patients receiving acupuncture had a lower risk of hypertension than those in the control group (adjusted SHR = 0.56, 95% CI = 0.54–0.58) after adjusting for potential confounding factors.

### 3.4. The Interaction EFFECT of Medication and Acupuncture on Hypertension

In the interaction analysis, Table 5 shows that patients who received medications, acupuncture, or combined therapy, tended to have a lower risk of hypertension compared to the nontherapy group. The group taking medications combined with acupuncture showed the lowest risk of hypertension (adjusted HR = 0.10, 95% CI = 0.09–0.12, *p*-value < 0.001).

## 4. Discussion

To our knowledge, this is the first study to demonstrate that the risk of hypertension in patients with CSU can be decreased by acupuncture, using data from the NHIRD with Cox regression models. The main findings of our study are as follows: (1) the benefits of acupuncture for reducing the incidence of hypertension in patients with CSU were independent of sex, age, baseline comorbidities, and drug use, and (2) the group taking medication combined with acupuncture had the lowest risk of hypertension.

The association between hypertension and CSU may be attributed to alterations in the endothelium and blood coagulation/fibrinolysis pathways [6]. Recent studies have reported that systemic hypertension is commonly associated with a procoagulant state [19]. Other studies have suggested that the coagulation cascade pathway may contribute to the pathogenesis of CSU. A recent study by Yuhki et al. suggested that medications targeting activated coagulation factors and/or complement components may represent new and effective treatments for patients with severe and refractory CSU [20]. A double-blind placebo-controlled study by Parslew et al. showed a response of CSU to warfarin [21].

Moreover, the association between hypertension and CSU may be related to inflammation [6]. Recent studies have found an association between inflammation markers and hypertension, and a study by Wang et al. found that biomarkers such as C-reactive protein (CRP) and plasminogen activator inhibitor-1 were significantly related to incident hypertension [22]. Studies have suggested that inflammation markers such as pentraxin-3 and CRP may be useful for the assessment of CSU disease activity [23,24].

In addition, hypertension and CSU may be linked to autoimmunity [6]. Mathis et al. suggested that the presence of self-antigens and the subsequent activation of the adaptive immune system against them may promote the development of hypertension [25]. Several studies have reported that oral cyclosporine and omalizumab are effective in treating patients with CSU, and the results provide further evidence supporting the hypothesis of an autoimmune pathogenesis of CSU [26,27,28].

The mechanism of the protective effect of acupuncture against hypertension in patients with CSU may involve the following three pathways: first, a study by Elisabet et al. found that low-frequency electroacupuncture counteracted a possible prothrombotic state in women with polycystic ovary syndrome [29], suggesting that the improvement of coagulation/fibrinolysis is a possible mechanism; second, the World Health Organization recommends acupuncture for the treatment of 16 types of inflammatory diseases [30], and several clinical studies have investigated whether acupuncture could treat inflammatory diseases and decrease CRP levels [31,32], suggesting an anti-inflammatory mechanism; and third, a study by Wang et al. suggested that acupuncture therapy has had promising results in a variety of autoimmune diseases, including multiple sclerosis, irritable bowel disease, rheumatoid arthritis, myasthenia gravis, systemic lupus erythematosus, and ankylosing spondylitis [33]. Another study by Sun Kwang Kim suggested that acupuncture enhances natural killer cell activity and can have dual immunomodulatory effects on either Th1- or Th2-skewed conditions to maintain homeostasis [34], altogether indicating an immunoregulatory mechanism.

Mental health disorders are associated with both CSU and hypertension. Several studies have suggested that anxiety and depression are associated with a higher risk of hypertension [35,36,37,38]. A study by Hamam suggested that physicians treating patients with hypertension should be aware of the role that anxiety and depression play in treatment efficacy and compliance [39]. Several studies have revealed that patients with CSU frequently have anxiety and depression, which are probably exacerbated by the symptoms of itching and sleep disturbance [40,41,42,43]. Studies have suggested that acupuncture treatment can alleviate anxiety and depression [44,45,46,47], which may benefit patients with CSU by decreasing the risk of hypertension.

One strength of our study is the use of population-based data that were highly representative of the general population. As more than 99% of patients in Taiwan are enrolled in the NHIRD, the external validity of the findings could be enhanced. However, the present results should be interpreted in light of the limitations arising from the nature of the retrospective analysis and characteristics of the NHIRD database. The evidence derived from retrospective cohort studies is generally of lower statistical quality than that derived from randomized trials because of potential bias related to the adjustments for confounding variables. Information on patient factors was not recorded in the NHIRD, and many important potential confounders were not controlled for in this analysis, which may result in residual confounding factors, such as family history, body mass index, diet, physical activity, smoking, alcohol consumption, and hypnotic medications. In addition, people using acupuncture may require more management by medications, and they may have an especially unique approach to treatment or may be more conscious of their health issues and management; therefore, self-selection bias may occur. Primary and secondary hypertension were not separated into groups in our study, which could also have led to some bias. In addition, the NHIRD database did not reveal the acupoints for CSU treatment; therefore, the mechanism of action of acupoints remains unclear.

Although the definite relationship between CSU and hypertension remains unclear, we discuss several possible pathways by which acupuncture decreases the risk of hypertension in patients with CSU. Further prospective studies using single or several acupoints, according to TCM theories or modern medical mechanisms, with or without Western medications, to investigate their relationships, may uncover clinically applicable strategies. These may be more effective in helping patients with CSU become aware of hypertension and receive a better diagnosis and treatment in advance, which will help improve disease prevention.

## 5. Conclusions

Our study found that acupuncture decreases the risk of hypertension in patients with CSU in Taiwan. Among the different therapy groups, medication combined with acupuncture was associated with the lowest risk of hypertension. This suggests that patients with CSU may choose acupuncture as an adjuvant therapy combined with regular medications for CSU symptoms to prevent the incidence of hypertension. The detailed mechanisms require further clarification in future prospective studies.

## Figures and Tables

**Figure 1 healthcare-11-01510-f001:**
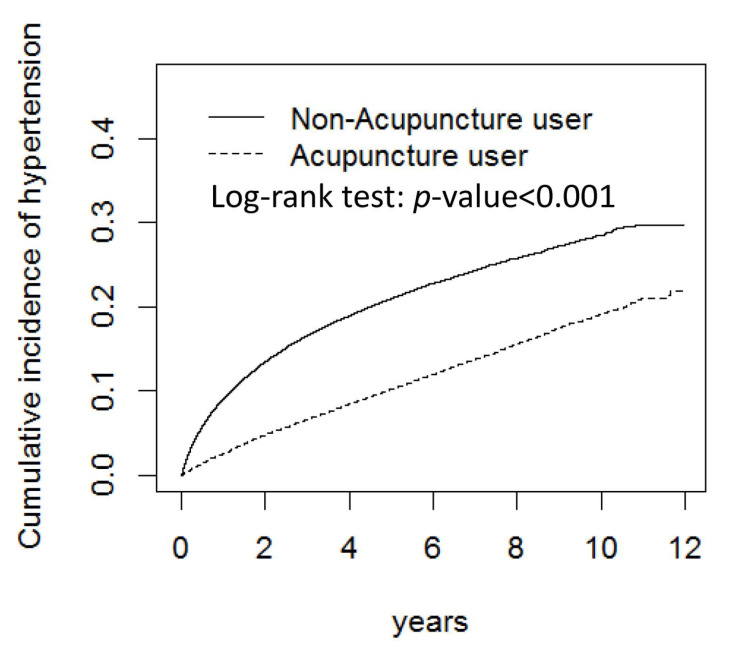
The cumulative incidence of hypertension during the follow-up period between the acupuncture and no-acupuncture groups.

**Table 1 healthcare-11-01510-t001:** Demographic characteristics of the non-acupuncture and acupuncture groups in patients. with chronic spontaneous urticaria.

Variable	Acupuncture				*p*-Value	SMD
	No (N = 43,547)		Yes (N = 43,547)			
	n	%	n	%		
Sex					0.44	
Female	29,762	68.3	29,868	68.6		0.005
Male	13,785	31.7	13,679	31.4		0.005
Age mean ± SD (years)	40.2 ± 13.2		40.1 ± 13.0		0.43 ^a^	0.005
Age group					0.21	
20–39	33,192	76.2	33,268	76.4		0.004
40–65	8282	19.0	8315	19.1		0.002
≥65	2073	4.76	1964	4.51		0.012
Urbanization level ^‡^					0.74	
1	24,502	56.3	24,399	56.0		0.005
2	16,604	38.1	16,638	38.2		0.002
3	1902	4.37	1950	4.48		0.005
4+	539	1.24	560	1.29		0.004
Monthly income (NTD) ^†^					0.84	
<20,000	6311	14.5	6360	14.6		0.003
20,001–39,999	24,435	56.1	24,356	55.9		0.004
≥40,000	12,801	29.4	12,831	29.5		0.002
Occupation					0.99	
White-collar	26,290	60.4	26,297	60.4		0.001
Blue-collar	11,073	25.4	11,059	25.4		0.001
Others ^&^	6184	14.2	6191	14.2		0.001
Comorbidity						
Diabetes mellitus	5427	12.5	5493	12.6	0.50	0.005
Hyperlipidemia	9148	21.0	9309	21.4	0.18	0.009
Medication						
Cetirizine	34,502	79.2	37,481	86.1	<0.001	0.181
Desloratadine	15,688	36.0	18,393	42.2	<0.001	0.138
Fexofenadine	38,815	89.1	40,811	93.7	<0.001	0.164
Levocetirizine	27,441	63.0	31,142	71.5	<0.001	0.182
Loratadine	28,415	65.3	31,355	72.0	<0.001	0.146
Follow time, years (mean, median)	(5.26, 2.85)		(3.50, 4.99)			0.615

*p*-value from Chi-square test, ^a^ *p*-value from independent *t*-test. SMD, standardized mean differences; SD, standard deviation. ^†^ New Taiwan Dollar (NTD). ^‡^ The urbanization level was divided by the population density of the residential area into four levels. ^&^ This included primarily retired, unemployed, and low-income populations.

**Table 2 healthcare-11-01510-t002:** Cox model with hazard ratio and 95% confidence interval of hypertension with receiving acupuncture and other covariates among the patients with chronic spontaneous urticaria.

Variable	Event	1000 Person-Years	IR	Crude		Adjusted ^$^	
	N = 12,559			HR (95% CI)	*p*-Value	HR (95% CI)	*p*-Value
Acupuncture							
No	7593	152,567	49.8	1 (Reference)		1 (Reference)	
Yes	4966	228,993	21.7	0.47 (0.46, 0.49)	<0.001	0.58 (0.55, 0.60)	<0.001
Sex							
Female	7517	265,328	28.3	1 (Reference)		1 (Reference)	
Male	5.042	116,232	43.4	1.52 (1.46, 1.57)	<0.001	1.22 (1.18, 1.27)	<0.001
Age (years)							
20–39	6278	306,833	20.5	1 (Reference)		1 (Reference)	
40–65	4466	63,047	70.8	3.32 (3.20, 3.45)	<0.001	2.32 (2.23, 2.42)	<0.001
≥65	1815	11,680	155.4	6.86 (6.51, 7.23)	<0.001	3.78 (3.57, 4.00)	<0.001
Urbanization level ^‡^							
1	6659	215,987	30.8	1.16 (0.97, 1.38)	0.10	1.16 (0.97, 1.38)	0.10
2	5047	144,753	34.9	1.31 (1.10, 1.56)	0.003	1.23 (1.03, 1.46)	0.02
3	723	15,944	45.4	1.68 (1.39, 2.02)	<0.001	1.31 (1.09, 1.58)	0.004
4+	130	4877	26.7	1 (Reference)		1 (Reference)	
Monthly income (NTD) ^†^							
<20,000	2342	53,213	44.0	1.57 (1.49, 1.66)	<0.001	1.42 (1.13, 1.79)	0.001
20,001–39,999	7034	212,911	33.0	1.19 (1.14, 1.24)	<0.001	1.02 (0.97, 1.07)	0.40
≥40,000	3183	115,436	27.6	1 (Reference)		1 (Reference)	
Occupation							
White-collar	6127	234,346	26.2	1 (Reference)		1 (Reference)	
Blue-collar	4165	95,051	43.8	1.67 (1.60, 1.73)	<0.001	1.22 (1.17, 1.28)	<0.001
Others ^&^	2267	52,163	43.5	1.65 (1.57, 1.73)	<0.001	0.79 (0.62, 0.99)	0.04
Comorbidities							
Diabetes mellitus							
No	9810	335,730	29.2	1 (Reference)		1 (Reference)	
Yes	2749	45,830	60.0	2.04 (1.95, 2.12)	<0.001	1.15 (1.10, 1.21)	<0.001
Hyperlipidemia							
No	8077	303,649	26.6	1 (Reference)		1 (Reference)	
Yes	4482	77,911	57.5	2.14 (2.07, 2.22)	<0.001	1.51 (1.45, 1.58)	<0.001
Medications							
Cetirizine							
No	4307	56,319	76.5	1 (Reference)		1 (Reference)	
Yes	8252	325,241	25.4	0.34 (0.33, 0.36)	<0.001	0.62 (0.60, 0.65)	<0.001
Desloratadine							
No	9027	225,286	40.1	1 (Reference)		1 (Reference)	
Yes	3532	156,274	22.6	0.57 (0.55, 0.60)	<0.001	0.93 (0.89, 0.97)	<0.001
Fexofenadine							
No	3773	23,511	160.5	1 (Reference)		1 (Reference)	
Yes	8786	358,049	24.5	0.16 (0.16, 0.17)	<0.001	0.32 (0.31, 0.33)	<0.0001
Levocetirizine							
No	7370	115,600	63.8	1 (Reference)		1 (Reference)	
Yes	5189	265,961	19.5	0.31 (0.30, 0.32)	<0.001	0.50 (0.48, 0.52)	<0.001
Loratadine							
No	5942	109,516	54.3	1 (Reference)		1 (Reference)	
Yes	6617	272,044	24.3	0.46 (0.44, 0.48)	<0.001	0.77 (0.74, 0.80)	<0.001

CI, confidence interval; HR, hazard ratio; IR, incidence rate per 1000 person-years. ^$^ Multivariable analysis included sex; age; urbanization level; monthly income; occupation; comorbidities of diabetes mellitus and hyperlipidemia; and medications of cetirizine, desloratadine, fexofenadine, levocetirizine, and loratadine. ^†^ New Taiwan Dollar (NTD). ^‡^ The urbanization level was divided by the population density of the residential area into four levels, of which level 1 was the most urbanized and level 4 was the least urbanized. ^&^ Other occupation categories included those who were primarily retired, those who were unemployed, and low-income populations.

**Table 3 healthcare-11-01510-t003:** Incidence rates, hazard ratios, and confidence intervals of hypertension for patients with CSU who received and did not receive acupuncture, stratified by sex, age, comorbidities, and medications.

	Non-Acupuncture			Acupuncture						
Variable	Event	1000 Person-Years	IR	Event	1000 Person-Years	IR	Crude		Adjusted ^$^	
							HR (95% CI)	*p*-Value	HR (95% CI)	*p*-Value
Sex										
Female	4575	106,632	42.9	2942	158,697	18.5	0.47 (0.45, 0.49)	<0.001	0.52 (0.50, 0.55)	<0.001
Male	3018	45,935	65.7	2024	70,297	28.8	0.48 (0.45, 0.51)	<0.001	0.56 (0.53, 0.60)	<0.001
Age (years)										
20–39	3792	124,381	30.5	2486	182,451	13.6	0.47 (0.45, 0.50)	<0.001	0.57 (0.54, 0.60)	<0.001
40–65	2745	23,799	115.3	1721	39,248	43.9	0.42 (0.40, 0.45)	<0.001	0.51 (0.48, 0.54)	<0.001
≥65	1056	4386	240.8	759	7294	104.1	0.48 (0.44, 0.53)	<0.001	0.57 (0.52, 0.62)	<0.001
Comorbidity ^†^										
No	4434	115,548	38.4	2793	170,320	16.4	0.47 (0.45, 0.49)	<0.001	0.54 (0.52, 0.57)	<0.001
Yes	3159	37,019	85.3	2173	58,674	37.0	0.47 (0.44, 0.49)	<0.001	0.53 (0.50, 0.56)	<0.001
Medications										
Cetirizine										
No	2944	27,582	106.7	1363	28,737	47.4	0.51 (0.48, 0.54)	<0.001	0.54 (0.51, 0.58)	<0.001
Yes	4649	124,985	37.2	3603	200,256	18.0	0.51 (0.49, 0.53)	<0.001	0.54 (0.52, 0.56)	<0.001
Desloratadine										
No	5705	95,098	60.0	3322	130,188	25.5	0.47 (0.45, 0.49)	<0.001	0.54 (0.51, 0.56)	<0.001
Yes	1888	57,469	32.9	1644	98,806	16.6	0.53 (0.49, 0.56)	<0.001	0.54 (0.51, 0.58)	<0.001
Fexofenadine										
No	2542	11,938	212.9	1231	11,574	106.4	0.58 (0.54, 0.62)	<0.001	0.58 (0.54, 0.62)	<0.001
Yes	5051	140,629	35.9	3735	217,420	17.2	0.50 (0.48, 0.53)	<0.001	0.53 (0.51, 0.56)	<0.001
Levocetirizine										
No	4850	52,620	92.2	2520	62,980	40.0	0.49 (0.47, 0.52)	<0.001	0.53 (0.50, 0.55)	<0.001
Yes	2743	99,947	27.4	2446	166,013	14.7	0.55 (0.52, 0.58)	<0.001	0.56 (0.53, 0.59)	<0.001
Loratadine										
No	3925	47,882	80.5	2017	60,744	33.2	0.46 (0.44, 0.49)	<0.001	0.53 (0.50, 0.56)	<0.001
Yes	3668	103,795	35.3	2949	168,249	17.5	0.52 (0.50, 0.55)	<0.001	0.55 (0.52, 0.58)	<0.001

CI, confidence interval; HR, hazard ratio; IR, incidence rate per 1000 person-years. ^$^ Multivariable analysis included sex; age; urbanization level; monthly income; occupation; comorbidities of diabetes mellitus and hyperlipidemia; and medications of cetirizine, desloratadine, fexofenadine, levocetirizine, and loratadine. ^†^ This included diabetes mellitus and hyperlipidemia.

**Table 4 healthcare-11-01510-t004:** Accepted acupuncture cohort to no-acupuncture cohort subhazard ratio of hypertension estimated using competing-risk regression models.

Variable	Crude		Adjusted ^$^	
	SHR (95% CI)	*p*-Value	SHR (95% CI)	*p*-Value
Acupuncture				
No	1 (Reference)		1 (Reference)	
Yes	0.48 (0.46, 0.50)	<0.001	0.56 (0.54, 0.58)	<0.001

CI, confidence interval; SHR, subhazard ratio. ^$^ Multivariable analysis included sex; age; urbanization level; monthly income; occupation; comorbidities of diabetes mellitus and hyperlipidemia; and medications of cetirizine, desloratadine, fexofenadine, levocetirizine, and loratadine.

**Table 5 healthcare-11-01510-t005:** Incidence and hazard ratios of hypertension among different therapy groups.

Acupuncture	Medication	N	Event	1000 Person-Years	IR	Crude HR (95% CI)	Adjusted HR ^$^ (95% CI)
No	No	995	712	1719	414.2	1.00	1.00
No	Yes	42,552	6881	150,848	45.6	0.13 (0.12, 0.14) ***	0.21 (0.19, 0.24) ***
Yes	No	339	195	1053	185.1	0.52 (0.44, 0.62) ***	0.51 (0.42, 0.62) ***
Yes	Yes	43,208	4771	227,940	20.9	0.06 (0.06, 0.07) ***	0.10 (0.09, 0.12) ***

CI, confidence interval; HR, hazard ratio; IR, incidence rate per 1000 person-years. ^$^ Multivariable analysis included sex; age; urbanization level; monthly income; occupation; comorbidities of diabetes mellitus and hyperlipidemia; and medications. ***: *p* < 0.001, *p*-value for interaction: 0.69.

## Data Availability

The dataset used in this study was obtained from the Taiwan Ministry of Health and Welfare (MOHW). The Ministry of Health and Welfare must approve of the application to access these data. Any researcher interested in accessing this dataset can submit an application form to the Ministry of Health and Welfare to request access. Please contact the staff of the MOHW (Email: stcarolwu@mohw.gov.tw) for further assistance. Taiwan Ministry of Health and Welfare Address: No.488, Sec. 6, Zhongxiao E. Rd., Nangang Dist., Taipei City 115, Taiwan (R.O.C.). Phone: +886-2-8590-6848.

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
