# Peer review of "Acupuncture Decreases Risk of Hypertension in Patients with Chronic Spontaneous Urticaria in Taiwan: A Nationwide Study"

_healthcare, 2023, doi:10.3390/healthcare11101510_

Round 1

Reviewer 1 Report

The objective of this study was to investigate whether acupuncture could decrease the risk of hypertension in patients with CSU. The results showed that medication combined with acupuncture had the lowest risk of hypertension among different therapy groups. This implies that patients with CSU may opt for acupuncture as adjuvant therapy and regular medications for CSU symptoms to prevent hypertension. This finding is significant in both Western and Chinese medicine.

This work is very interesting. I, therefore, recommend publication in the Healthcare Journal. I have one suggestion that the authors can address:

The authors could add one paragraph to the section “Introduction,” which includes brief descriptions of the methodology and results of this study. 

Author Response

Response to Reviewer 1 Comments

Point 1: The authors could add one paragraph to the section “Introduction,” which includes brief descriptions of the methodology and results of this study.

Response 1: Thank you for your kindly suggestion. We have added one paragraph to the section “Introduction” including brief descriptions of the methodology and results of our study to improve the quality of the manuscript.

Thank you very much.

Line 71: The study was conducted between 2008 and 2019 and included patients with CSU who received acupuncture or not. The primary endpoint was the occurrence of hypertension. The two cohorts were matched through 1:1 propensity score matching by age, sex, index year, comorbidities, medications, monthly income, urbanization level, and occupation. The study included 43,547 patients with CSU receiving acupuncture matched with 43,547 patients with CSU who did not receive acupuncture. The results showed that the acupuncture group had a significantly lower risk of hypertension than those in the control group (adjusted HR = 0.58, 95% confidence interval [CI] = 0.55–0.60). In the interaction analysis, patients receiving medications combined with acupuncture tended to have the lowest risk of hypertension.

Reviewer 2 Report

The paper by authors from Taiwan described an interesting topic, a nationwide study of the influence of acupuncture on the occurrence of hypertension among patients suffered from chronic spontaneous urticaria (CSU).

The paper is clearly written.

 This study shows that patients with CSU treated with acupuncture have a significantly lower risk of hypertension than those without acupuncture treatment, regardless of sex, age, comorbidities and medications. The lowest risk of hypertension have patients taking medications combined with acupuncture.

The influence of acupuncture on the mechanisms that are considered to be involved in the development of hypertension and CSU is well-discussed.

The conclusion is directly related to the results.

The paper deserves to be published.

However, I have just two minor comments.

1. In table 1, in the last line, the authors should state that the follow up period was measured in years - Follow time, years (mean, median).

2. The Discussion section should start emphasizing the main findings of the study. For example: “The main findings of our study are 1)…, 2)…,3)…”

Author Response

Response to Reviewer 2 Comments

Point 1: In table 1, in the last line, the authors should state that the follow up period was measured in years - Follow time, years (mean, median)..

Response 1: Thank you for your kindly suggestions. We have added the measure unit (years). Thank you very much to help us improve the quality of the article.

Point 2: The Discussion section should start emphasizing the main findings of the study. For example: “The main findings of our study are 1)…, 2)…,3)…

Response 2: Thank you for your kindly suggestions for us to emphasize more the main findings of the study. It will make the finding of our study more clear for readers. Thank you very much.

Line 222: The main findings of our study are 1) The benefits of acupuncture for reducing the incidence of hypertension in patients with CSU were independent of sex, age, baseline comorbidities and drug use, 2) The group taking medication combined with acupuncture had the lowest risk of hypertension.

Reviewer 3 Report

Dear Editor, Thank you for giving me the opportunity to revise this paper.

In this case-control study, the authors investigated the potential benefit of acupuncture in decreasing the risk of hypertension among patients with chronic spontaneous urticaria (CSU). They used 2 cohorts of 43,547 patients with CSU, matched by a propensity score matching (PSM) procedure and gave acupuncture treatment to one cohort only. The incidence of hypertension over time was lower among patients taking medications and acupuncture, followed by those taking acupuncture and those not taking acupuncture. 

The manuscript is well written, and supported by references. I have only a few minor concerns:

-Diagnosis of hypertension should be better described. Authors specified ICD-9 codes but it is not clear how they have been used. Did authors use discharge codes for patients hospitalized during the follow-up time? Or did they assess BP directly during follow-up visits?

-In Table 3, it is not clear which is the reference level for the HRs; for instance in each medication, you specified a decreased risk for HPT for the level "yes" and "no", that should not be possible. Check this typo and additionally add the note for the superscript $.

English language is fine and there are only some minor typos.

Author Response

Response to Reviewer 3 Comments

Point 1: Diagnosis of hypertension should be better described. Authors specified ICD-9 codes but it is not clear how they have been used. Did authors use discharge codes for patients hospitalized during the follow-up time? Or did they assess BP directly during follow-up visits?

Response 1: Thank you for your kindly suggestions. In this study, we use The Taiwanese National Health Insurance Research Database (NHIRD) and the healthcare information includes outpatient visits, hospital admissions, and prescription medications. And we used diagnostic codes from the International Classification of Disease, Ninth & Tenth Revision, Clinical Modification (ICD-9-CM & ICD-10-CM) to detect all patients with hypertension. At least two times from principal/secondary diagnoses in outpatient visits or 1 visit with hospitalizations from 2008 to 2019 during follow-up period were needed for hypertension diagnosis.

BP of inpatients or outpatients would be assessed by medical members and the diagnosis would be made by the visiting doctor using ICD-9-CM or ICD-10-CM codes and we obtained data from NHIRD.

Line 106: Hypertension patients were identified at least two times from principal/secondary diagnoses in outpatient visits or hospitalizations from 2008 to 2019 during follow-up period.

Point 2: In Table 3, it is not clear which is the reference level for the HRs; for instance in each medication, you specified a decreased risk for HPT for the level "yes" and "no", that should not be possible. Check this typo and additionally add the note for the superscript $.

Response 2: Thank you for your kindly suggestions. There are a variety of drugs to choose in clinical practice, so doctors may not necessarily use every drug, thus, there will be two options of Cetirizine (and other medications) yes or no. And the reference level for the HRs is subjects with non-acupuncture. For example, among subjects with “no” Cetirizine treatment, subject with acupuncture (IR = 47.4/1000 person-year) had lower risk of hypertension (HR = 0.51(0.48, 0.54)) compared to those subjects with non-acupuncture (IR = 106.7/1000 person-year).

And we have double checked for each table and make sure that the descriptions of symbols are included in the footnote.

Reviewer 4 Report

Thank you for the opportunity to review this paper. This paper fits perfectly my main expertise in applied statistics and the assessment of population data. The paper presents a population cohort study evaluating the effectiveness of acupuncture treatment for managing patients with hypertension with chronic spontaneous urticaria. The paper overall is very well written with language and a narrative that is easy to follow and understand. The study uses the appropriate methods to do the assessments.

I would be happy to recommend this paper for publication with some brief clarifications and addressing of some concerns that I believe will improve the quality of the paper. Please allow me to elaborate.

Detailed comments:

First, I am not completely sure on why this is about hypertension with chronic spontaneous urticaria, why not just hypertension. The chronic spontaneous urticaria management was not assessed as an outcome. I wonder if you would make a clearer point on why hypertension with chronic spontaneous urticaria and not just hypertension. I apologize if I do not see something obvious.

The only concern I have with the paper is that propensity score matching was used for age, sex, index year, comorbidities, medications, income, urbanization level and occupation, BUT medication usage was constantly and significantly different between the two main groups. This suggests self-selection bias may be happening. People using acupuncture may require more management by medications, they may have an especially unique approach to treatment or may be more conscious of their health issues and management. This may be a caveat that should be added to the discussion.

Table 2 and table 3 have the “$” in the Adjusted but it is not defined in the footnote. Table 5 does define it.

Last, what was the percentage of data missingness mentioned in line 88?

Great work, looking forward to your reply.

Author Response

Response to Reviewer 4 Comments

Point 1: First, I am not completely sure on why this is about hypertension with chronic spontaneous urticaria, why not just hypertension. The chronic spontaneous urticaria management was not assessed as an outcome. I wonder if you would make a clearer point on why hypertension with chronic spontaneous urticaria and not just hypertension. I apologize if I do not see something obvious.

Response 1: Thank you for your kindly comments. Our previous study found that the CSU was associated with a higher future risk of hypertension (line 49, reference [6]) , so the study aim is to determine whether acupuncture could decrease the risk of hypertension in patients with CSU and these may help improve hypertension prevention.

Point 2: The only concern I have with the paper is that propensity score matching was used for age, sex, index year, comorbidities, medications, income, urbanization level and occupation, BUT medication usage was constantly and significantly different between the two main groups. This suggests self-selection bias may be happening. People using acupuncture may require more management by medications, they may have an especially unique approach to treatment or may be more conscious of their health issues and management. This may be a caveat that should be added to the discussion.

Response 2: Thank you for your kindly suggestions. In our study, we performed propensity score matching to construct a comparative cohort for acupuncture users from the study population. Propensity scores (PS) were calculated as a probability dependent on a vector of observed covariates associated with receipt of treatment with acupuncture. We conducted a logistic regression analysis for estimating PS with adjustment for age, sex, index year, comorbidities, medications, income, urbanization level and occupation. A 1:1 PS-matched cohort was constructed using greedy nearest neighbor matching, and the caliper width was within 0.1. Moreover, standardized mean differences (SMD) with a cutoff value of 0.10 were used to observe the fitness of covariate comparisons between the propensity score matched groups. (ref1-3) Therefore, we have added the information of SMD for each variable on Table 1.

So medications were included in the PS matching but SMDs are more than 0.10 which indicated that medication usage was still constantly and significantly different between the two main groups (Table 1). Therefore, in our following analyses, we did add medications as covariate in the adjusted Cox models. More, table 3 shows that regardless of medication usage, acupuncture users tended to have a lower risk of hypertension than non-acupuncture users (adjusted HRs < 1, p <0.001).

We also added the possible self-selection bias to the discussion according to your suggestion. Exactly, people using acupuncture as an adjuvant therapy may have an especially unique approach to treatment or may be more conscious of their health issues and management.

Line 281: Besides, people using acupuncture may require more management by medications, and they may have an especially unique approach to treatment or may be more conscious of their health issues and management, therefore, self-selection bias may be happening.

Ref1: Performing a 1:N Case-Control Match on Propensity Score. SUGI29, 2001:165-29

(The link: http://www2.sas.com/proceedings/sugi29/165-29.pdf)

Ref2: One-to-many propensity score matching in cohort studies. Pharmacoepidemiol Drug Saf. 2012 May;21 Suppl 2:69-80. doi: 10.1002/pds.3263.

Ref3: Austin PC. Optimal caliper widths for propensity-score matching when estimating differences in means and differences in proportions in observational studies. Pharm Stat. 2011;10(2):150-161. doi:10.1002/pst.433

Point 3: Table 2 and table 3 have the “$” in the Adjusted but it is not defined in the footnote. Table 5 does define it.

Response 3: Thank you for your kindly suggestions. We have double checked for each table and make sure that the descriptions of symbols are included in the footnote.

Point 4: Last, what was the percentage of data missingness mentioned in line 88?

Response 4: Thank you for the question. A total of 1.33% (2879/216173) of CSU patients were with missing sex and age data. A flow chart of study design is added as a supplementary material (Figure S1) for clarifying.
